# Synthetic Biology for Terraformation Lessons from Mars, Earth, and the Microbiome

**DOI:** 10.3390/life10020014

**Published:** 2020-02-09

**Authors:** Nuria Conde-Pueyo, Blai Vidiella, Josep Sardanyés, Miguel Berdugo, Fernando T. Maestre, Victor de Lorenzo, Ricard Solé

**Affiliations:** 1ICREA-Complex Systems Lab, Universitat Pompeu Fabra, Plaça de la Mercè, 10, 08002 Barcelona, Spain; blai.vidiella@upf.edu (B.V.); mglberdugo@gmail.com (M.B.); 2Institut de Biologia Evolutiva, UPF-CSIC, 08003 Barcelona, Spain; 3Centre de Recerca Matemàtica, Campus UAB Edifici C, 08193 Bellaterra, Barcelona, Spain; jsardanyes@crm.cat; 4Barcelona Graduate School of Mathematics (BGSMath), Campus UAB Edifici C, 08193 Bellaterra, Barcelona, Spain; 5Departamento de Ecología and Instituto Multidisciplinar para el Estudio del Medio “Ramon Margalef”, Universidad de Alicante, Carr. de San Vicente del Raspeig, s/n, 03690 San Vicente del Raspeig, Alicante, Spain; ft.maestre@gmail.com; 6Systems and Synthetic Biology Program, Centro Nacional de Biotecnología (CNB-CSIC), Campus de Cantoblanco, 28049 Madrid, Spain; vdlorenzo@cnb.csic.es; 7Santa Fe Institute, 1399 Hyde Park Road, Santa Fe, NM 87501, USA

**Keywords:** terraformation, Mars, evolution, microbiome, synthetic biology, drylands, hypercycles, restoration ecology

## Abstract

What is the potential for synthetic biology as a way of engineering, on a large scale, complex ecosystems? Can it be used to change endangered ecological communities and rescue them to prevent their collapse? What are the best strategies for such ecological engineering paths to succeed? Is it possible to create stable, diverse synthetic ecosystems capable of persisting in closed environments? Can synthetic communities be created to thrive on planets different from ours? These and other questions pervade major future developments within synthetic biology. The goal of engineering ecosystems is plagued with all kinds of technological, scientific and ethic problems. In this paper, we consider the requirements for terraformation, i.e., for changing a given environment to make it hospitable to some given class of life forms. Although the standard use of this term involved strategies for planetary terraformation, it has been recently suggested that this approach could be applied to a very different context: ecological communities within our own planet. As discussed here, this includes multiple scales, from the gut microbiome to the entire biosphere.

## 1. Introduction

Our biosphere is the result of a long term evolutionary process spanning billions of years. A very diverse range of ecological communities are present in all habitats known, from rainforests to drylands or the Antarctica, but organized in similar ways across the planet and respond to challenges in predictable ways [1]. Among the future scenarios of biodiversity change, catastrophic shifts appear to be a likely outcome of global warming [2,3,4,5]. Avoiding ecological meltdown will require major efforts and a plethora of strategies [6]. Given the continuous growth of the human population and the massive use of fossil sources of energy (among others [7]), along with an over-exploitation of natural resources, a planetary tipping point is likely to be reached [2,3,4]. However, these efforts might also require the development of novel technical solutions to counterbalance the accelerated pace of changes and a window of opportunity that rapidly shrinks. In this context, climate models and ongoing evidence from a plethora of field studies clearly indicate that runaway effects will be unleashed [8]. For instance, the Sahara desert once was vegetated and rapidly became the desert that it is at present [9,10].

Among other possible strategies aiming to counterbalance climate change impacts, very different strategies have been suggested, from sustainable energy and growth policies to geoengineering. The last involves (among other things) technological solutions aimed at reducing the impacts of greenhouse gases [11,12,13,14]. Geoengineering is aimed at operating directly on diverse physical or chemical factors. Among other proposed solutions, we can mention carbon dioxide capture and storage, ocean iron fertilisation, or even using billions of free-flying spacecrafts to cool down the planet [11,14,15]. These potential approximations can involve staggering costs. Some of them are cooling approaches (i.e., negative radiative forcing) aimed at changing Earth’s albedo, thus not directly acting on the CO${}_{2}$ excess [16]. Lower costs would be achieved by the production and deployment of aerosols [17].

A rather different proposal offer a strategy that implies bioengineering the biosphere [18,19,20]. In a nutshell, the core idea is that to counterbalance the impact of warming and its associated tipping points, engineered organisms would be designed and deployed to directly act on the reduction of greenhouse gases, removal of undesirable waste or as a way to enhance given properties required for habitat persistence. One key argument in favor of this strategy is that the “machines” required to achieve the desired goals are alive and thus self-replicate themselves. Because of the engineering perspective associated with this approach and since it seeks the preservation of a habitable planet (particularly to humans) the term *Terraformation* was used. This was originally used within the context of the anthropogenic, large-scale transformation of planets, with Mars as the main case study [21,22,23,24]. The original proposal included a planetary-scale process that would push Mars freezing temperatures and thin atmosphere to a more Earth-like state where liquid water could be stable. Such goal, despite all the technological limitations, would be favored by the observation that Mars is currently very close to the triple point (in the pressure-temperature phase space) where liquid water can exist.

What lessons can be gathered from early work on Mars terraformation that can be relevant to the Earth’s bioengineering? And the other way around: what relevant clues can be obtained from our understanding of complex communities on Earth to guide future strategies aimed at changing Mars geochemistry and bring living organisms there? In this paper, we seek to explore these and other related questions by making a critical comparison between the Mars and Earth case studies along with a third one: the microbiome (shown in Figure 1a–c respectively). This is a conceptually fundamental finding within biology: the fact that multicellular systems host a community of microorganisms interacting and coevolving with them in such a way that a coherent, higher-order living organization is at work. This is the so-called *Holobiont* [25,26] and dedicated work on the origins, evolution and ecology of these communities include both a better understanding of living systems [27] but also strategies to modify them [28,29]. Such possibility is particularly relevant to restore damaged microbiomes affected by diseases [30] or environmental stress [31]. Mounting evidence indicates that appropriate interventions can shift the ecological networks to novel (not necessarily natural) states [30,32,33]. Since microbiomes are themselves large, complex ecosystems, they provide a virtually infinite number of potential terraformation-like experiments.

The engineering of synthetic ecosystems requires new ways of dealing with modified organisms beyond the single-species level. In fact, this is a major challenge while developing a theoretical framework, whee multiple layers are relevant, as sketched in Figure 2. As we move from genetic circuits to the organism and community levels, the uncertainties about the impact of genetic designs increase. Dealing with the effects of deployment of engineered microorganisms is inextricably tied with our capacity of understanding multilevel ecological complexity.

Synthetic biology combines and re-designs available genetic elements and re-purposes them into new molecular circuits to create microbes with defined properties to fulfill specific desired goals (Figure 2c,d). Considerable advances in genetic engineering have been developed in the last two decades [34,35,36]. Different methods can be used to systematically assemble DNA fragments in a modular fashion. Examples are Golden Gate, Gibson assembly [37], MoClo [38] including CRISPR/Cas systems for efficient gene deletions, insertions, and transcriptional control [39] and their cyanobacteria counterpart [40] (particularly relevant for our discussion). Some have been adapted to non-standard organisms, including Loop assembly [41] or CyanoGate [42]. Moreover, by the use of evolutionary engineering, we can obtain organisms adapted to new environments or exhibiting new desired phenotypes [43,44,45].

Presently, we can successfully engineer cells to sense exogenous inputs [46,47], and control gene expression [48,49,50,51] and communication [52,53,54]. Synthetic interactions within a microbial consortia are typically engineered using the natural quorum sensing system for bacteria [55,56], or different pheromones in fungi [57,58]; but adhesion proteins or ion channels can be also used [59,60]. To integrate the behavior of an individual within a consortium, different genetic circuits for population control [61,62], distribution of tasks [63,64], dynamic coordination [61,65], and spatial organization [66,67,68], have already been built. Further extensions allow designing and program multi-species synthetic ecosystems [66,69,70,71]. Microbial interactions (i.e., competition, cooperation, and mutualism) have been successfully engineered by tinkering with syntrophic exchanges [72,73,74,75]. Using all these genetic tools, the first steps towards ecosystem engineering were made (i.e., within the levels shown in Figure 2a,b). An example of this capability is the gene drive strategy to reduce the malaria mosquito vector abundance [76] and its improved version by means of CRISPR-Cas9 [77]. These are just first steps towards a more ambitious program full of challenges, including strategies to make it work, both in terms of efficient spreading and genetic functional tasks [78]. To quickly propagate genetic constructs through the entire environmental microbiome (as part of the multiscale synthesis), much can be learned of horizontal gene transfer mechanisms (i.e., antibiotic resistance genes [79]). For instance, its use to engineere gene-drive-like artificial devices for multiplying gene transfer events in the absence of a pressure [80,81,82,83]. For real-world applications, new approaches should tackle community robustness and stability and biocontainment measures.

In this paper, we mainly explore three classes of systems where terraformation strategies can be approached both theoretically and experimentally. Three images of these systems are displayed in Figure 1, including: (a) Mars, (b) drylands, and (c) the gut microbiome. The features associated with each of them (with a wide range of spatial and temporal variability) are considerably different. However, the goal of modifying Mars to create an Earth-like biosphere, the repair and recovery of our damaged planet and the engineering of microbiomes to fight disease exhibit deep connections. Specifically, we will focus on microbiome-based bioengineering approaches considering the following topics:**Terraformation**: How and why two major approaches to terraformation, namely geoengineering and bioengineering, might be crucial to modify or generate ecosystems in the three case studies defined above. We will discuss how bioengineering can be a particularly efficient strategy and why synthetic biology can be the appropriate technological approximation. The differences between Mars and Earth terraformation goals and strategies will be outlined.**Engineering drylands and synthetic soils**: A crucial problem affecting the three case studies described above (Mars, degraded ecosystems and damaged microbiomes) is the need to push these systems to reliable states where living communities can maintain their desired diversity and properly manage external fluctuations. Taking drylands as a case study, and particularly abrupt soil disruption reported at certain aridity levels, we will see how a multi-scale approximation to these communities is required in order to efficiently design intervention scenarios grounded in using species of microorganisms present in the soil microbiomes. Within this context, we present the concept of *synthetic soil* as a keystone in ecosystem redesign.**Ecological hypercycles**: So-called terraformation motifs, i.e., specific engineered networks of interacting species, have been proposed as design principles for terraformation using synthetic biology. However, to achieve landscape-level or even planetary-level targets, special classes of dynamical interactions, the so-called ecological hypercycles (defined as closed cooperative consortia), are required. We discuss the rationale for these dynamical designs and why they might be crucial for successful interventions.**Microbial hypercycles**: The potential impact of synthetic terraformation might be of great importance in future missions aimed at modifying (locally or globally) the climate and soil properties. How can ecological hypercycles be designed for planetary missions? This includes in particular several challenges associated with closed ecosystems and how to create the appropriate communities that can meet those associated with the presence of human microbiomes, as well as the design of proper ecosystems and required substrates for their persistence.

The nature and implications of the previous topics and problems will be discussed at the end of the paper, emphasizing the diverse connections between them, the potential caveats of each approach and a basic outline of a roadmap for a theory of ecosystem terraformation.

## 2. Planetary Terraformation: From Geo-To Bioengineering

One of the original proposals concerning Mars terraformation was suggested by James Lovelock [84]. A once watery and temperate planet that is presently bitterly cold and dry, but that could be changed, using Greenhouse gases, such as perfluorocarbons, which are capable—in principle—of warming Mars and increasing the density of its atmosphere which would in turn made liquid water stable. Several major issues emerge when evaluating the potential success of this strategy. Once this warming has taken place, changes would mainly take place through a long-term transformation triggered by a biotic community able to amplify the initial deviation from the current equilibrium state. How can we explore these issues?

In an idealized setting, a terraformed Mars would experience a set of transitions from a dry, polar desert to a planet displaying green landscapes and even perhaps grasslands. Such a linear succession sequence has more to do with a crude extrapolation from Earth’s communities than a truly scientific approach to what is actually possible. It is too often ignored that the there was a terraformation event (among others) that made possible the invasion of land and that is the evolution of soils linked to a complex mixture of plants and microbes [85,86]. Plants in particular became ecosystem engineers [87,88,89], i.e., crucial elements to control energy and matter flows in expanding habitats. Soils became the fabric of biocomplexity, plants started to develop a whole set of novel structures (the root systems) in the dark side while microorganisms coevolved with them [90]. As it occurs with other traits of our biosphere, the coevolution between biological and environmental properties pervades the creation of habitats suitable for the maintenance of complex and diverse life forms.

Mars terraformation (Sagan 1973, article “Planetary engineering on Mars” [91]) would likely require the combination of both strategies. In 1973, Carl Sagan [91] suggested that a sustained transport of $10^{2}$–$10^{3}$ metric tons of some low-albedo material to the ice cap regions of Mars could efficiently change Mars climate (over the course of a century). He also suggested to use a “dark plant” capable of having the same impact: growing on the polar snows, it would accomplish the same objective. Sagan already indicated that none of the two scenarios was likely to happen in the “near future”. Almost fifty years later, both scenarios remain far from realistic as originally formulated. However, an enormous knowledge leap in planetary science, and particularly in our understanding of Mars, has taken place. Water presence in the underground and multiple sources of geological evidence reveal a planet that used to be wet and a global picture of the planetary limits and potentials is emerging. Similarly, recent decades have witnessed the development of synthetic biology as an engineering avenue for modifying cells to perform novel functionalities [92]. This area promises the modification of living systems in ways that could overcome the design limitations resulting from evolutionary trade-offs [93,94]. Could engineered microorganisms be the key for planetary terraforming?

To approach the previous questions, it is helpful to consider the idea of biosphere terraformation outlined above. The suggestion of using synthetic biology to engineer Earth’s habitats [18] implies a starting point that is rather different from the Mars engineering. In a nutshell, terraforming Mars is largely a bottom-up design. It will require locating the environment under a favourable set of conditions that will define the boundary conditions for potential life forms. These living systems, likely to be limited to microorganisms, will be introduced with the aim of taking advantage of the geoengineering process and push it forward in ways similar to those that changed our primitive planet. Once in place, microorganisms should not only thrive in the Red Planet (while dealing with all kinds of radiation or water shortage issues) but also change its climate.

Using Lovelock’s toy model *Daisyworld* [95,96,97,98], engineering at both the climate and microorganisms levels should provide an initial state from which the Martian ecology would start evolving, but starting from a near-equilibrium state. Instead, the proposed biosphere terraformation scheme would be a top-down one. Our biosphere is already made of networks of interacting species under a range of favourable conditions. Modifications of habitats or communities necessarily start from an organized structure that is far from equilibrium. Interventions based on synthetic biology trigger changes in well-established biological communities, exploiting the presence of multiple equilibrium states [18,20,30,66,75]. The potential avenues opened by this idea are wide, but given the context analysed here, we will consider an especially relevant case study, namely the potential for terraformation of arid and semiarid ecosystems.

## 3. Terraforming Drylands: Synthetic Soils

In its original formulation [18] it was suggested that some specific case studies could be the target of the terraformation approach for living ecosystems. As it occurred with our neighboring planets, unless something is done, ours might also experience a runaway effect. In particular, Earth’s climate will rapidly move towards a high-CO${}_{2}$ [16], high-temperature levels, with little chances for survival of the vast majority of species. Semiarid and arid ecosystems are especially relevant to our discussion, since they are likely to be among the first to move across tipping points. They are the vastest biome on Earth, covering more than 45% of emerged lands [33,99,100,101] and hosting a third of human populations [102,103].

This class of ecosystem is characterized by water stress and extreme temperature changes, with vegetation patterns that vary over a wide range of possibilities (but is typically patchy) [104,105,106,107]. The extreme case of this repertoire is provided by deserts, perhaps the context closest to the Mars problem. Because of these features, some particular habitats such as the Atacama desert have been used as a source of understanding of what can be expected for Mars. This habitat is extremely dry, with rains that might not occur for centuries, with specialized microbial communities that have been dormant for extended periods of time. Other arid locations that are considered the Earth case studies closest to Mars include Dry Valleys in Antarctica, Death Valley in California or Devion Island in Canada (among others) [23]. Because the extreme conditions strongly limit the possibilities of invader species to succeed, appropriate strategies will need to consider the use of extremophiles [108,109] or the engineering of new microbial life forms able to cope with those conditions. Synthetic biology is likely to be the most promising way of dealing with the gap to restart a novel biosphere.

Two main questions can be formulated in relation to our two main terraformation scenarios. The Mars-related one is: How can a bioengineering strategy help pushing Mars into a new stable state allowing life to colonize the planet and co-evolve with a warmer and wetter climate? The answer requires to remember that the starting point of Mars from a geological point of view is a wet planet that experienced a runaway effect towards a new dry, cold and carbon dioxide-dominated stable configuration. Within the context of the warming biosphere, the question now is: How can we use synthetic biology (along with other climate engineering strategies) to prevent our planet from experiencing a runaway effect once tipping points are reached? The asymmetry here is compelling: Earth might face, as Mars did, a nonlinear process of accelerated change towards a warm climate where complex ecosystems will experience a massive extinction event [2,7,110].

Although a very diverse range of potential bioengineering strategies can be imagined, a first roadmap towards a systematic design has been introduced under the label of *Terraformation motifs* [18,19,20]. These motifs describe the minimal interaction networks that would be required to engineer endangered ecosystems or act as a bioremediation strategy to modify human-generated ecosystems associated with waste. This would include for example sewers or plastic wastelands. A systematic list with four major classes where presented in [19] and the population dynamics of these motifs was analysed in [20]. One especially important case scenario was the problem of tipping points in semiarid ecosystems, which we will explore in detail (including the role of the space, see ref. [111]).

At the individual level, drylands are inhabited by organisms that have adapted to low moisture, damaging radiation levels and extreme temperatures. At the plant community level, dryland ecosystems developed a complex and highly relevant way of interaction between species, with non-trophic interactions, notably, facilitation, playing a determinant role in amelioration microclimatic conditions at local scales [112]. These interactions suppose the emergence of positive feedbacks in the ecosystem that are associated with processes of self-organization [113] and heterogeneity generation [114] that expands ecosystem development beyond natural limits of vegetation [115]. Those interactions are acknowledge to be responsible for the emergence of catastrophic shifts under increasing environmental constraint in modelled ecosystems [106] scaling up even to the ecosystem level and resulting in desertification [116].

Because tipping points deeply modify the concept of risk (there is no linear decay response as some external variables change) a different approach is needed to deal with their nature. Finally, also at the ecosystem level, different studies conducted in the field suggest the incidence of abrupt transitions in the soil system along spatial gradients of increasing aridity [117]. In particular, at certain aridity levels, corresponding to the inter-fase between semiarid and arid ecosystems, some studies have reported an abrupt transition of soil functioning (soil fertility and nutrient turnover rates) which correspond to patterns that fit in a catastrophic shift behavior [118] (see Figure 3). Those findings have been reported in several ecosystems, globally [118] and regionally [119], and attain nitrogen cycling rates [119,120], micronutrients [121], soil fauna [122,123], changes in the relative importance of erosive agents [124], soil aggregates stability and the relationship between soils and plants [117]. Presumably, those changes would involve also a drastic change in soil microorganisms [125,126], which are the main biotic agents driving and connecting soil nutrients and stocks [127,128] and contribute importantly to enhance soil stability and water capacity.

Such accumulation of proofs around the same aridity levels focus the attention on the vulnerability of dryland soils to aridity increases that may attain a drastic soil disruption under ongoing climate change [130]. At the least, if these abrupt changes in soil functioning scale through community mechanisms (e.g., facilitation and micro-environment creation), they may promote vast community changes, e.g., shrub invasions in a phenomenon called shrub encroachment [131]. If scaling to the composition level affecting diversity, and surpassing species capacity to adapt to the new conditions, they may even cause desertification and transitions to desert-type landscapes. Those provide the closest context to the Mars terraformation problem. Whatever the case, the abrupt nature of reported soil losses suggest that these changes may also be stable in time involving tipping points that may not be recovered unless specific actions change the underlying dynamical mechanisms of such shifts.

Three lessons of interest for terraformation approaches in our own planet can be extracted from the literature mentioned above. First, the importance of soils as a key compartment in drylands. Its disruption by increasing aridity causes large losses of multiple ecosystem services. This is likely linked with the development of microbial soil communities which are keystone ecosystem engineers [132], and, therefore, this calls to focus effort for terraformation on soil microorganisms. Second, positive biotic feedbacks, notably facilitation, are a natural mechanism for self-sustaining of drylands soils that can be exploited in a terraformation canvas for engineering degraded drylands. In addition, third, because the extreme conditions strongly limit the possibilities of invader species to succeed, appropriate strategies will need to consider the use of extremophiles [108,109] or the engineering of new microbial life forms able to cope with those conditions. Synthetic biology of soil microbial life forms is likely to be the most promising way of dealing with the gap to restart a novel biosphere [18,133].

Let us consider a specific case study and how to develop a mathematical model that can give us some insight into the mechanisms of terraformation based on synthetic biology. The soil crust can be described as complex living skin spanning a few centimeters of the topsoil [134,135]. These are remarkable communities hosting a wide variety of species and largely mediating the energy and matter flows through the soil surface. In general, the more arid the environment the less diverse the community, and, since plants and the biocrust are strongly related to each other, increased aridity leads to a smaller vegetation cover, less organic carbon, reduced plant productivity, and loss of multifunctionality [127,136,137].

If we reduce our interest (as it is done in basic population dynamics) to a small subset of species, a terraformation motif can be described in terms of low-dimensional system, thus ignoring the multi-species nature of these communities. Two basic schemes representing the interactions between the different components of the motif are shown in Figure 4b,c. The first case involves a direct impact through some tight relationship with the host plant (Figure 4b), which can be, for example, an engineered symbiosis [138] or an enhaced previously existing symbiont from the rhizosphere community [139,140,141,142,143]. The second case (Figure 4c) relies on an indirect cooperation mediated by the influence of the $S_{w}$ species on e.g., moisture. Let us consider and analyse the two scenarios separately.

A reasonable model for this motif is a system of coupled differential equations as follows:(1)$$\begin{array}{ccc} \frac{dH}{dt} & = & \Gamma \left(H,S\right)\left(1-\frac{H}{K}\right)-\epsilon H \end{array}$$(2)$$\begin{array}{ccc} \frac{dS}{dt} & = & \left(\eta H+\rho )S-\mu S-S\Phi (S\right) \end{array}$$(3)$$\begin{array}{ccc} \frac{dS_{w}}{dt} & = & \mu S+\rho_{w}S_{w}-S_{w}\Phi \left(S\right) \end{array}$$
describing the interactions between two strains (SYN with state variable *S*; and WT with state variable $S_{w}$) competing for available resources (or space) while the engineered SYN strain is involved in a cooperative interaction with the host. Here Γ(H,S) is a growth function and *K* the carrying capacity of the vegetation cover. The parameter ϵ is the density-independent host death rate of the host while η is the growth of *S* due to the mutualistic interaction with the host. The function Φ(S) also stands for the outflow of the system, introducing competition. As we previously did, under the CP assumption, we can reduce the microbial pair dynamics to a ingle equation with $\Phi \left(S\right)=\eta HS+\rho S+\rho_{w}S_{w}$ (see [20]) and thus the equation for the synthetic population reads now:$$\frac{dS}{dt}=\left(\eta H+\rho -\rho_{w}\right)S\left(1-S\right)-\mu S.$$
Here Γ(H,S)=(r+γS)H, assuming that, in general, the host is capable of growing (at a rate *r*) in the absence of the microbial strains whereas the term γHS stands for the cooperative interaction. A detailed analysis of this (and other classes of terraformation motifs) was analysed in detail in Solé, et al. 2018 [20]. The model reveals a rich landscape of possible dynamical states, including sharp transitions between them.

An important question in general is how changes in the efficiency of the cooperative interaction help the system reach a stable state where both plants and microorganisms coexist. If the rate of recovery of the wild type is high, the cooperative interaction between synthetic and plant might become weak whereas an efficient growth of the synthetic can make mutualistic exchanges stable over time. In Figure 4d we show an example of the extinction (white) versus coexistence (green) phases associated with the failure and success of the synthetic motif. Similarly, using the parameter space (γ,μ) (i.e., facilitation of the synthetic from plants and restoration rate of the wild type), the surface in Figure 4d corresponds to the population abundance of the synthetic strain, which rapidly decays to extinction as we cross a well-defined critical line.

## 4. Ecological Hypercycles: Design Principles for Terraformation

What is the most efficient terraformation motif that can help developing the desired functionality? What makes the previous motif a good candidate to efficiently propagate the synthetic strain across the ecosystem? Which architectures of cooperative interactions among motifs ensure long-term persistence of the entire system? These are relevant questions for all the case studies explored here. In this section, we show that one specific design principle is embedded in the previous example and that it is a requirement for propagating populations by exploiting a cooperative loop. The key is a mutualistic cycle where different species help others to replicate [145,146], using a closed set of catalytic interactions known as the hypercycle.

The original hypercycle model [147] describes the dynamics of a set of replicators with a closed catalytic architecture (Figure 5a and Figure 6b). The hypercycle model was conceived within the framework of origins of life to investigate the dynamics of information coded by replicators with catalytic properties (e.g., ribozymes). The generality of the hypercycle model describing cooperative dynamics have been used to investigate other systems with nonlinear feedbacks such as ecosystems [148,149,150]. A general model for hypercycles with *n* species including both Malthusian and catalytic growth reads:(4)$$\frac{dx_{i}}{dt}=k_{i}x_{i}+k_{i,j}x_{i}x_{j}-x_{i}\Phi \left(\vec{x}\right).$$
Here $x_{i}$ is the population amount of species *i* (with i=1,…,n), $x_{j}$ being the previous species in the catalytic network. Constants $k_{i}$ and $k_{i,j}$ denote the exponential self-replication of species *i* and the catalytically-assisted replication that species *j* provide to species *i*, respectively. Since the architecture is cyclic, we shall impose j=n when i=1 and j=i−1 when i≠1. The first right hand side (rhs) in Equation (Equation 4) is the Malthusian growth, which follows exponential dynamics, and considers that the hypercycle species can reproduce by themselves (i.e., without needing catalytic aid). The second rhs term defines the non-linear growth (heterocataoysis) that a given member provides to the next one in the cycle. When only the catalytic growth is considered one talks about the obligate hypercycle. The last term, $\Phi (\vec{x})$, is the dilution outflow that keeps population constant and introduces competition between the hypercycle members.

The dynamics of hypercycles have been thoroughly investigated during the last 40 years. Hypercycles have two interesting properties: the growth of species is faster than exponential (see below), and the cyclic architecture ensures the persistence of all members since the competitive exclusion principle does not apply in hypercycles. It is well known that hypercycles are bistable systems and their persistence state depends on key parameters such as the decay of the species. That is, there exist critical decay values above which hypercycles are not able to persist. Moreover, this critical value involves a discontinuous transition (similar to the one displayed in Figure 3c–e and Figure 6b, see below). The dynamics of hypercycles largely depends on the number of species (see Figure 5d for cases n=3,4,5). For example, the persistence dynamics of two-species systems are governed by a stable equilibrium [147,151,152,153]. Three- and four-species systems also coexist by stable equilibria, although these equilibria are achieved with transient oscillations [147,154,155]. Hypercycles with more than five species typically undergo self-sustained periodic oscillations [156,157]. Research on hypercycles goes beyond the theoretical and computational models. In recent decades, several experimental hypercycles have been built at different levels: at the polypeptide level with coiled-coil proteins [158] and at the population level with two yeast strains cooperating in the production of essential amino acids [74]. Also, an autocatalytic system, in which the individuals of one specie provide catalytic aid to themselves, have been recently build with yeast [159] (see Figure 6a). More recently, a synthetic two-member bacterial hypercycle with a catalytic parasite has been also investigated [75].

Let us introduce here one of the simplest models of a hypercycle, which involves a pair of species helping each other [152,153]. The population densities of each member of the cooperative loop, will be given by $x_{1}$ and $x_{2}$, respectively. The equations for this system would read:(5)$$\begin{array}{c} \frac{dx_{1}}{dt}=\Gamma_{12}x_{1}x_{2}\left(1-\frac{x_{1}+x_{2}}{K}\right)-\delta_{1}x_{1}, \end{array}$$(6)$$\begin{array}{c} \frac{dx_{2}}{dt}=\Gamma_{21}x_{2}x_{1}\left(1-\frac{x_{1}+x_{2}}{K}\right)-\delta_{2}x_{2}. \end{array}$$
The coefficients $\Gamma_{ij}$ stand for the replication rate of each species under the presence of the second. Here, instead of using the constant population constraint (as in Equation (Equation 4)) we include competition and boundedness of solutions with a logistic function, *K* being the carrying capacity. The last term stands for linear degradation rates. For simplicity we consider the symmetric case, where $\Gamma_{12}=\Gamma_{21}\equiv \Gamma$ and $\delta_{1}=\delta_{2}\equiv \delta$ (the asymmetric case does not involve a qualitative change of the dynamics, see ref. [152]). Under these assumptions, it can be shown that at equilibrium $x_{1}^{*}=x_{2}^{*}=x$. The dynamics can in fact be reduced to a single differential equation model for *x* [161]:(7)$$\begin{array}{c} \frac{dx}{dt}=\Gamma x^{2}\left(1-\frac{2x}{K}\right)-\delta x. \end{array}$$
The first property to highlight about this system is the presence of the so-called hyperbolic growth [147]. Consider the previous system for a small population (i. e x≪K) and such that decay rate is very small. In this case the previous equation can be approached to
(8)$$\begin{array}{c} f\left(x\right)=\frac{dx}{dt}\approx \Gamma x^{2} \end{array}$$
which can be easily solved, leading to the solution:(9)$$\begin{array}{c} x\left(t\right)=\frac{x(0)}{1-x(0)\;\Gamma \;t}. \end{array}$$
Please note that as a difference from exponential growth (obtained from dx/dt=Γx, with solution x(t)=x(0)exp(Γt), hyperbolic kinetics can reach infinite concentration of *x* in finite times. This growth kinetics, compared with the exponential one, is shown in Figure 5b. Notice that while exponential growth proceeds in a monotonous manner, the hyperbolic one involves very small populations at the beginning. However, the system suddenly explodes to infinity once the condition x(0)Γt=1 is achieved (in Figure 5b we have used x(0)=01 and Γ=0.5 and thus the previous condition is fulfilled at t=20). That is, The presence of such accelerated growth implies that a hyper-exponential growth of the population is expected when approaching the divergence time. Importantly, these populations can overcome exponentially-growing populations.

The equilibrium points of Equation (Equation 7), obtained from dx/dt=0, are $x_{0}=0$ and the pair
(10)$$x_{\pm }=\frac{K}{4}\left(1\pm \sqrt{1-\frac{\Gamma_{c}}{\Gamma }}\right),$$
where we use $\Gamma_{c}=8\delta /K$. It is easy to show that the equilibrium $x_{0}$ is always stable. The analyses of $x_{\pm }$ reveal that for $\Gamma_{c}=\Gamma$, $x_{+}=x_{-}$, meaning that these two equilibria collide. When $\Gamma >\Gamma_{c}$ the two equilibrium points exist and are different. Actually it can be shown that $x_{+}$ is stable while $x_{-}$ is unstable. When $\Gamma <\Gamma_{c}$ the term inside the square root of equilibria (Equation 10) becomes negative and the equilibria do not exist. The collision of these two equilibria at $\Gamma_{c}=\Gamma$ causes a catastrophic transition (i.e., saddle-node bifurcation [162]). That is, below the transition the two hypercycle members persist, while after the transition they become extinct. This can be observed in the bifurcation diagram of Figure 5c.

Another way to visualise the equilibrium dynamics is by means of the computation of the so-called potential function, computed from:(11)$$U\left(x\right)=-\int f\left(x\right)\;dx=-x^{2}\left(\Gamma x\left(\frac{1}{3}-\frac{x}{2K}\right)-\frac{\delta }{2}\right).$$
This function has been plotted tuning δ for different values of *x*. This gives the surface displayed in Figure 5e, where the different regions marked with arrows denote the states of extinction (grey arrows) and persistence (red arrows).

The previous model is a very simple one, and might appear as an oversimplification when compared with the previous equations associated with the terraformation motifs described above. However, it can be shown that both models share the same potential for fast and stable hyperbolic growth. Consider now a simpler parameter set and an early initial condition where both populations are far from saturation (i.e., H≪1,S≪1). If we assume obligate mutualism (i.e., if r=0,ρ=0), normalize K=1 and assume that the engineered strain is not likely to shift back to the wild type (i.e., μ≈0) it is possible to show that the equations reduce to
(12)$$\begin{array}{c} \frac{dH}{dt}=\gamma HS, \end{array}$$
(13)$$\begin{array}{c} \frac{dS}{dt}=\eta HS, \end{array}$$
which captures the essential of the cooperative loop: both populations will grow, under a multiplicative kinetics, provided that both are present. The rates of growth are thus the same up to a constant, i.e., dH/dt=(γ/η)(dS/dt), since both components of this consortium will be present altogether. Let us assume that one follows the other under a linear relation, i.e., S(t)=ψH(t). In this case, it is easy to show that the growth of *H* follows:(14)$$\frac{dH}{dt}=\frac{\theta \psi }{1+\psi }H^{2}$$
which is precisely the quadratic model described for the simple hypercycle. This convergence with the simple counterpart of the hypercycle suggests that a fast acceleration in the population spread will take place.

The proposed chain of positive interactions between the synthetic organisms and their host is the same as proposing a new or an enhanced positive feedback loop. When both species interact with positive interactions, this can be also seen as an hypercycle. This kind of positive interaction chains are frequent in nature. Indeed, they are the key process of nutrient recycling that humans have disrupted in the last centuries [5,163,164] and now are disrupting the whole biosphere. As discussed in Wilkinson 2007 [165] this kind of catalytic loop was first mentioned in Charles Darwin’s work on earth worms [166]. Darwin realized that earth worms act as ecosystem engineers, by improving the quality of soils which help supporting vegetation cover, with plants contributing back to further soil formation where more worms can live. Cooperative loops are actually central to understand or engineering explosive dynamical phenomena, and it seems reasonable to suggest that they would be very efficient for spreading the impact of terraforming motifs in arid ecosystems or invading microbiomes. Concerning the Mars target, these results suggest that in order to use a designed or evolved microorganism capable of spreading at the planetary scale, a cooperative consortium exhibiting cooperative links would be much more reliable to get there. In the next section, we offer some clues about design principles to achieve these goals.

## 5. Synthetic Microbiomes: Hypercycles for Engineered Ecosystems

In previous sections, microbial communities have been shown to be crucial for understanding the collective dynamics and resilience of soil crusts in drylands. The models described above support the concept that an efficient propagation of the terraformation designs requires, as a necessary condition, a hyperbolic-like dynamics. Beyond the kinetic description and the robust mathematical properties displayed by hypercycles, the next step requires getting closer to biology.

All biological macro-molecules are mainly formed by a small subset of atoms, namely H, C, N, O, S, and P. These are the main building blocks of planetary biogeochemical cycles: a set of nested abiotic acid-based and biotic red-ox reactions, have evolved to require low external energy. The biological fluxes of these elements are carried out by reversible metabolic pathways of synergistic cooperation of multi-species assemblages [167,168]. In this context, engineering closed catalytic populations might be essential to all the scenarios discussed here. To restore the earth biochemical cycles, disturbed by industrial anthropocentric activities, and to re-create them on Mars environment or enclosed spatial stations. Any (re)engineered ecosystem design must aim at the circularization of all biochemical resources. In other words, nutrient and mater cycling must not be short-circuited by metabolic products that cannot be biologically reused.

An idealized picture of these cycles can be envisioned by considering them as the reaction networks of a closed ecosystem. In a closed ecosystem, no matter exchange is allowed, while we allow light intake and heat loss. This is also valid for artificial biospheres of all kinds, including small ecospheres [169], space stations [170,171] or large projects such as Biosphere 2 [172,173]. A successful design would include the creation of self-sustaining ecosystems. It is still challenging to predict the number and kind of species required to guarantee the persistence of the ecosystem [73], or at least its ecological services [174]. This is also a requirement for synthetic designs and for the establishment of novel communities in non-Earth environments (including planets). When designing an enclosed microbial ecosystem, there are fundamental properties that, must be ensured, including the maintenance of population abundances and diversity (with the help of cross-colonization between different habitats), the presence metabolic persistence and robustness against cheaters.

Although community diversity is seldom considered when designing synthetic ecosystems, distributing the reaction pathway among different microbial members minimizes the metabolic burden, and increase the productivity in the synthetic ecosystems inside bioreactors. A diverse population of microorganisms with a minimal population size is needed to perform the right transformation. The population equilibrium of the different bacteria within a consortium has to be the adequate to maintain stochiometry. Division of labor in microbial communities also facilitate optimal operation since each step of the pathway can enjoy the specialized intracellular conditions of the most suitable host microorganism. Although the chosen species may not grow in similar conditions, they can be made grow in separated but interconnected environments. This spatial compartmentalization allows having specialized environments that maximize the function of the engineered microbiome while decreasing undesired side products.

Synthetic biology advances promise to allow designing microbial consortia with specialized tasks. In that context, how to divide the tasks can be understood as a “multicellular distributed computation” problem. It has been shown that designed architectures, non based in engineering/electronic standards approaches, permit taking advantage of the inherent cellular modularity, gaining scalability and resilience [57]. By means of this engineering approach, division of labour can be easily achieved and controlled. This is particularly easy and flexible when compartmentalization is included [63] although a general set of design principles to different case studies (from bioreactors or urban areas) needs to be developed.

Going back to the advantages of hypercyclic networks to propagate synthetic designs, one can ask how the autocatalytic loops described above can be implemented forming closed networks of reactions. At the microbial level, commensal relations, where one organism feeds on the metabolic waste of another, are very common. Auxotrophs reduce the metabolic burden while promoting cooperation, since the bacteria rely on extracellular sources of amino acids for survival. Engineering such mutualistic symbiosis is already a reality, as shown in the examples of Figure 6. This is predicted to be feasible in co-cultured strains of engineered E. coli forming a closed cycle Figure 7. As a rule of thumb, we can predict that autocatalytic systems provide both a source of rapid spread as well as tight dependencies that guarantee the maintenance of diverse consortia. Since these mutualistic relations may be only maintained in specific conditions, it would be advantageous a design where the same metabolic step must be assigned to multiple microorganisms. Each faction of same-metabolic-step species, will also share the same engineered cooperative relation with the species of the factions that perform other metabolic steps (see Figure 7b).

In terms of the modelling of these catalytic cycles associated with auxotrophic interactions, several models can be defined. The simplest example is the following two-equation description where we limit the level of definition to microbial species abundance ($S_{i}$) along with the concentration of amino acids ($aa_{i}$) required to build the closed mutualistic loop: (15)$$\begin{array}{ccc} \frac{d}{dt}S_{i} & = & \alpha_{i}\;aa_{i-1}\;S_{i}\left(1\;-\;\sum_{j}^{N_{S}}\;S_{j}\right)\;-\;\delta \;S_{i} \end{array}$$(16)$$\begin{array}{ccc} \frac{d}{dt}aa_{i} & = & \rho_{i}\;S_{i}\;-\;\gamma_{i}\;aa_{i}\;S_{i+1}-\;\delta \;aa_{i} \end{array}$$
where $N_{S}$ is the total number of species and $N_{F}$ are the organisms in the same functional group *F*. The variables are: $\alpha_{i}$ is the growth rate of $S_{i}$, $\rho_{i}$ production rate of the i-th amino acid $aa_{i}$ due to $S_{i}$, $\gamma_{i}$ consumption rate of $aa_{i}$ by $S_{i+1}$ and δ is the dilution rate. The difference between panels c and d in Figure 7 is the limit resource constrain. It is modeled by the logistic term $\left(1\;-\;\sum_{j=1}^{N_{S}}\;S_{j}\right)$ when considering the spatial spread over a given area (Figure 7a) while it is just controlled by dilution and cooperation rates (Figure 7b).

In the previous sections we have been considering the bioengineering of three main types of systems. The examples are dominated by what we known about the biosphere, with both small-scale (the microbiome) and large-scale (the planet) scenarios. As a third case study, we also consider the potential of synthetic biology to engineer a novel biosphere in a planet devoid of life. In the three examples there is an underlying set of genetic designs whose chances of success we largely ignore. Engineering new species means to redefine the existing network of interactions so that we can restore previous steady states or perhaps create novel ones. Our discussion concerning hypercycles and cooperative designs in general suggests that any promising design would require building new symbiotic relations.

Beyond these examples (but closely related to the underlying problem of tipping points) new designed microorganisms might play a key role in dealing with the problems derived from industrial metabolism [175]. In all these cases, engineering needs to guarantee the proper propagation of the engineered agents [176]. The current mainstream thoughts of the safe use of such agents have focused on their containment by means of genetic firewalls to prevent undesirable, unpredictable dissemination of the genetically engineered organisms and/or they engineered DNA [177,178]. Yet, the need for scaling up might require thinking on entirely different terms, looking for large-scale spreading of the synthetic circuits [78]. The technical challenge in this case is not so much containment but just the contrary: massive spreading. The knowledge of such highways of DNA dissemination [79,179] and their drivers might become important for planning the spreading of good traits as well [80]. Modeling and validation of such promiscuous genetic designs remains an open problem.

## 6. Discussion

The original formulation of the terraformation concept involved the possibility of modifying the planetary-level conditions of Mars (or further planets as proposed in *Directed panspermia* [180,181]) to make its atmosphere and average temperatures closer to what is required for human life. This is a long shot dream, and many obstacles prevent such a possibility from becoming a reality. The concept however, was coopted in a different way to consider the possibility of terraforming our own planet [18]. In this case, our proposal aims at using synthetic biology to find potential ways out from the accelerated degradation of ecosystems resulting from global warming, as well as a strategy for dealing with the major sources of waste and contamination resulting from a non-circular industrial metabolism [175]. In stark contrast with the classical Mars scenario, we deal with a living planet where life has been thriving for billions of years, and thus interventions have to be planned in ways that include the population dynamics of the resident community.

Among the potential applications of synthetic biology, synthetic ecosystems define one of the most challenging ones. Using microorganisms to modify rich, diverse and complex living communities can be instrumental to interrogate natural systems on multiple scales, from microbiomes to the whole biosphere. Along with a better understanding of these systems and their synthetic (modified) counterparts, the potential for bioremediation, repair or terraformation allows considering novel scenarios to repair damaged communities. This is the case of microbiome-associated diseases as well as endangered ecosystems, such as drylands. Lessons from both domains (although they are not separated, as discussed above) can be exchanged as a better understanding of them emerges. In a more broad sense, both scenarios are close, since they both involve diverse, well-established networks of interactions, both trophic and mutualistic. Moreover, synthetic modification of ecological systems deals with much larger scales, the major role played by soil microbiomes reveals that the microbial-scale level is likely to play a key role.

In this paper, we used three major case scenarios, but they are actually part of a richer space of possibilities. In Figure 8a we summarize some relevant examples as located within a space that allows a qualitative comparison between them. The scale of the systems involved changes in many orders of magnitude along different axes, It involves three axes that make use of (a) the complexity of the measurable reaction networks associated with each example. These reaction networks are typically associated with the web of exchanges between different parts of the underlying metabolic web but also can be chemical reactions, as those taking place in interstellar space; (b) the number of species involved in each system is a second axis, where our planet would occupy the highest-richer limit, whereas Venus and (so far) Mars are located in the zero-species extreme. Intermediate cases include a few-species in in vitro synthetic ecosystems and a very diverse array of microbiomes but also Biosphere 2 (Figure 8b). The third axis (c) is associated with temperature. Here Venus is placed in one vertex of the space whereas Antarctic ecosystems (Figure 8c) would occupy the opposite vertex. Among other examples, the structure and organization of some microbiomes are worth mentioning. One example is the microbiome of solar panels, first analysed in Dorado-Morales, et al. 2016 [182]. This is a very interesting example of a community of extremophiles adapted to harsh environmental conditions. These communities appear to be closer to desert-like habitats than anything like a urban microbiome or an industrial reactor. Similarly, Biosphere 2 was a human-managed, unique experiment of dealing with a subset of the biosphere (involving several habitats, including a desert-like one [183]) changing under closed ecosystem conditions.

The space of possible ecosystems and planetary boundaries described above is a qualitative one. It locates the different examples in relative terms, since we do not attempt to provide quantitative measures. Nevertheless, it can be a useful guide for future efforts in defining a unifying picture of terraformation. It is a good picture of the diverse range of relevant case studies, their relative similarity and what intermediate problems might be found or designed. In this context, our paper suggests a few ingredients associated with the potential paths that could be taken. Diversity is an important dimension, since diverse ecosystems are more resistant to failure and thus might be more robust to deal with synthetic strains, while terraformation associated with Mars or other planetary candidates might consider species assemblies instead of single-species approximations. There is a physical context that might also require special attention. The relevance of soil microbiomes has been highlighted above in the arid/semiarid context. The soil crust is not only a perfect example of a rich system but also an inspiration for future designs of living communities on Mars, where any strategy might require some inspiration from drylands, to be adapted to the special conditions of the Martian soil. Finally, a no less important lesson from our comparative analysis concerns the nature of the interactions within the designed communities, where cooperative loops (as described by the ecological hypercycles) might be crucial for success.

The synthetic biology approach taken here is not free of uncertainties. A major concern is the potential derivatives associated with the release of engineered strains [184,185,186,187,188,189]. Such problem is presently being considered by researchers targeting microbial species belonging to the human skin or gut microbiomes. It was also a source of major controversy in the early 1980s and remains debated today as gene drive technologies start to be developed and used. Those early attempts of using microbes to treat crops rapidly became banned, thus effectively removing any possibility of testing their real impacts. What can be said about their potential interference with ecosystem-level functions? Some answers will be provided by future microbiome-related biomedical developments, since they might reveal how ecosystem-level changes can be triggered by engineering on the molecular scale. In general, we need to answer these questions and seriously consider what do we want. If humans are to be part of the future biosphere, technological solutions must be considered. The window of opportunity is getting narrower and Mars is not part of the solution. As discussed by most climate change researchers, despite the limitations of all these approximations, the price of not preparing for the future will be immense [190].

## Figures and Tables

**Figure 1 life-10-00014-f001:**
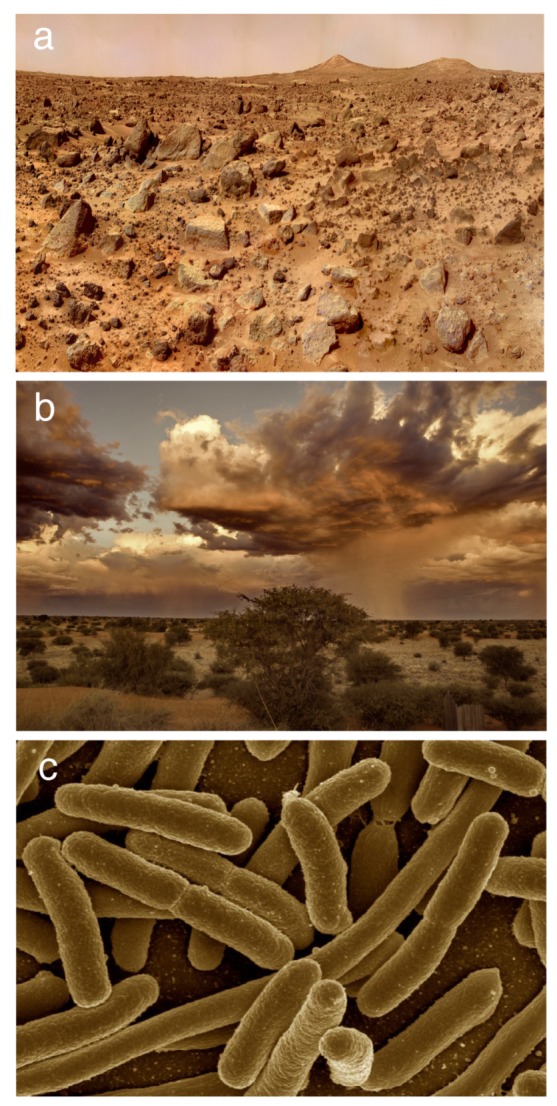
Terraformation: three different scenarios are depicted here as exemplified by their underlying landscapes. The classical use of the term was first applied to Mars, a planet nowadays likely to be devoid of life. A typical view is given in (**a**), corresponding to a Pathfinder image of a region named Twin Peaks (Image of NASA JPL). Earth’s landscapes instead are largely dominated by biodiversity, even in those ecosystems experiencing environmental stress, as it is the case of drylands. An example is given in (**b**) showing a semiarid land in the Kalahari Desert (image from Hanspeter Baumeler). An apparently different situation corresponds to the microbiome that can be found, for example, in the rich ecology of the gut microbiome, where E. coli (panel **c**, adapted from Rocky Mountain Laboratories, NIAID, NIH) is a well-known example.

**Figure 2 life-10-00014-f002:**
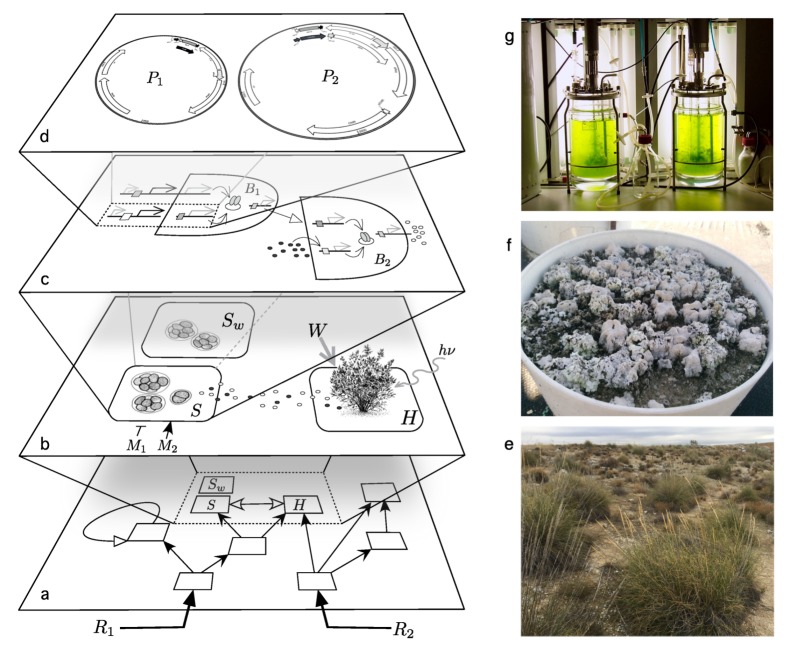
Terraforming the biosphere: Multiple levels are involved in the development of strategies for synthetic ecosystem terraformation based on synthetic biology (adapted from [20]). Several scales of complexity need to be considered. This includes (**a**) whole community dynamics. A mesoscale level (**b**) involving groups of a few relevant species, which can include synthetic candidates (here indicated as *S*) derived from a wild-type strain (here indicated as $S_{w}$) and a host *H* that in this case is a plant. At the cell level (**c**) we move into the definition and testing of cell circuits and their chassis. Finally, at the gene-sequence level (**d**) designed constructs need to be engineered to operate under predictable circumstances. In drylands (**e**) a key piece of the community architecture is provided by the soil crust, which can be experimentally manipulated (**f**) to test a diverse range of climate-related problems. Synthetic biology can help stabilize these communities, preventing them from crossing degradation thresholds or tipping points. One especially relevant candidate that can be easily cultivated in bioreactors (**g**) are cyanobacteria, which are known to play a crucial role in arid and semiarid ecosystems.

**Figure 3 life-10-00014-f003:**
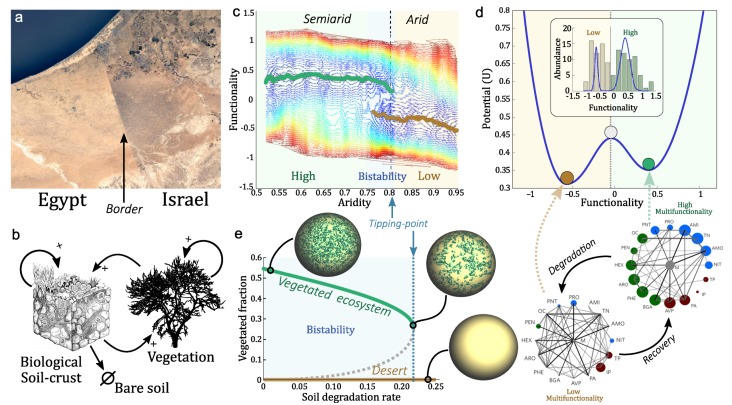
Bistability and tipping points in semiarid ecosystems. (**a**) Two diferent vegetated states coexist under the same environmental conditions, but different ploicies drive the ecosystems to diferent states (Egypt-Israel border from Google Earth Engine). (**b**) This bistability arises from the interactions between the vegetation and the sorrunding microbiome, the Biological soilcrust. The basic schema displayed in (**b**) is the one used in Kefi, et al. and used in many other studies [100,111,129]. Functionality in semiard ecosystems depends on the aridity (**c**). Depending on the aridity there are three regions: High functional ecosystems (low aridity, green region), Low functional ecosystems (high aridity, brown region), and a bistable region where both can coexist (intermediate aridity, blue region). From experimental data it can be observed that statistically there is only two possible configurations (**d**) and the other ones are transients between those two. The interactions between ecosystem functions change, not only their interactions strength (below). Figure adapted from Berdugo, et al. 2017 [118]. In panel (**e**), it can be seen the same bistability in the vegetation than in the other traits as functionality (adapted from [129]).

**Figure 4 life-10-00014-f004:**
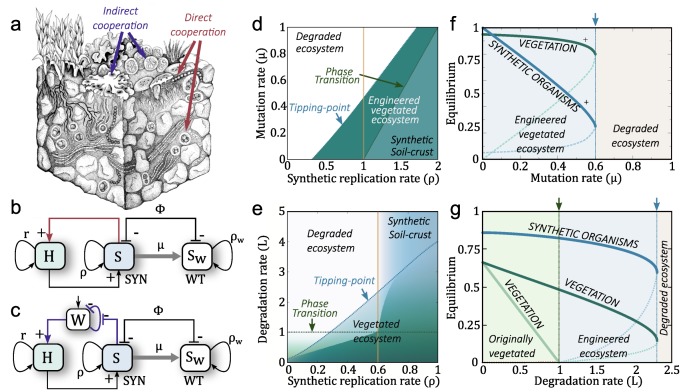
Engineering soil microbiomes in drylands using cooperative engineered interactions. The soil crust (**a**) is the ecosystem where the cooperative engineering will be performed (image from Belnap, et al. [144]). Here a cooperative loop involving cooperation among synthetic (SYN) microorganisms designed from a resident wild type (WT) and multicellular hosts (H) is considered. The SYN can revert to the WT either by losing the engineered construct at a rate μ. The two motifs shown correspond to direct (**b**) and indirect (**c**) positive interactions among both partners defining a mutualistic dependency. In panel d we summarise some results of the mathematical model associated with strict direct cooperation with r=0. Here (**d**) shows the regions in the parameter space (ρ,μ) where the host and the synthetic strains (SYN) survive (green area) and become extinct (white area), using γ=0.5. In (**e**) the stationary SYN strain populations are displayed for different combinations of (ρ,L) from the indirect cooperation motif. Notice the sharp transition taking place once a critical line (doted blue) is reached. This tipping point is enphasised by the blue arrow in panels (**f**,**g**). Moreover, the vegetation have a previuos phase transition when L=1 (green doted line and arrow). For more detailed information see Solé, et al. 2018 (ref. [20]).

**Figure 5 life-10-00014-f005:**
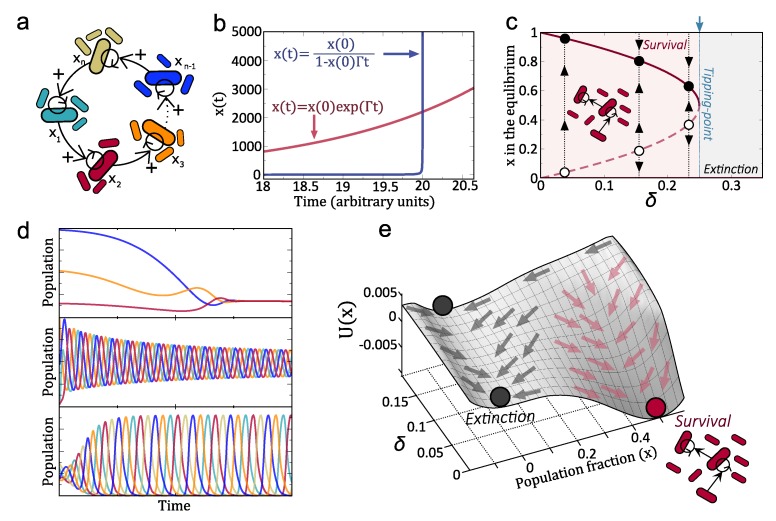
(**a**) Hypercycle with *n* species. (**b**) Comparison between exponential (red line) and hyperbolic (blue line) growth kinetics. Here we have used x(0)=01 and Γ=0.5. (**c**) Bifurcation diagram increasing δ. The hypercycle species always coexist due to cooperation in its persistence regime (with $\Gamma <\Gamma_{c}$, see Equation (Equation 10)). (**d**) Persistence dynamics for hypercycles with n=3 (upper with fast damped oscillations); n=4 (middle with hardly damped oscillations); and n=5 (lower with sustained periodic oscillations) members. (**e**) Potential function tuning δ obtained from Equation (Equation 11). The grey arrows indicate the scenarios where the species extinct, while the red arrows show the region with hypercycle persistence.

**Figure 6 life-10-00014-f006:**
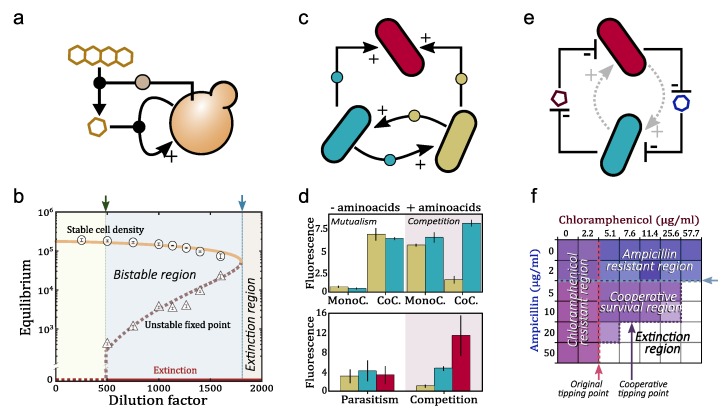
Some examples of minimal synthetic hypercycles. (**a**) The minimal hypercycle is the autocatalytic one. In the case of Dai, et al. 2012 [159], yeast cooperate in order to obtain nutrients. When the dilution rate of yeast is high enough, the population suffers a tipping point (panel **b**). Note this tipping point is the same as the one shown in Figure 5c, governed by a saddle-node bifurcation. (**c**) Hypercycle built with two bacteria strains and a parasitic one [75]. In this “synthetic ecosystem”, depending on the availability of amino-acids in the culture, the system reaches different population equilibria (**d**), they can either compete or cooperate. (**e**) Cooperation between species can be also stablished via inhibition of threads that affect the other species [160]. The cooperative loop used for degrading antibiotics allows the cooperative system to persist under higher antibiotic concentrations (**f**) than those tolerable for single components (purple dashed line), even if they are already resistant (blue and red dashed lines).

**Figure 7 life-10-00014-f007:**
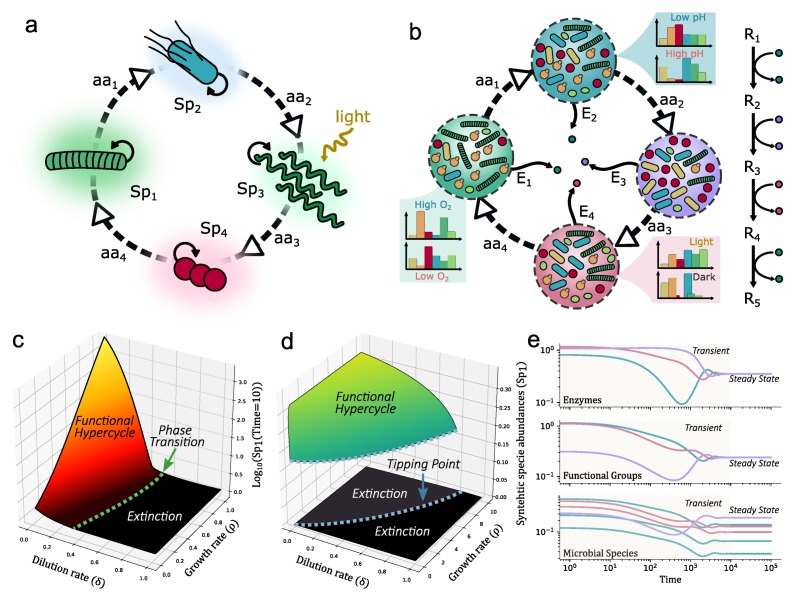
Synthetic hypercycles for terraformation. Taking the strict cooperation motif from [20], and making a chain of them, we obtain a hypercycle that ensures the presence of all the species (**a**). However, in a Martian context, we may want to ensure the production of certain metabolites in order to maintain a circularity of the overall metabolism (**b**). In this motif, the same function would be implemented by different species, achieving a more resilient consortium. The closed autocatalysis guarantees that all the functional groups will be maintained but the species fraction could change depending on external factors (here qualitatively illustrated by means of histograms). In a Martian context, the autocatalytic cycle can be useful in two situations. First, if the synthetic organisms are deployed on the planet, depending on their death and growth rates, they will grow explosively or get extinct. This can occur in different ways (**c**,**d**) depending on the type of dynamical regime (**c**: spatial spread, **d**: bioreactor) considered. The bioreactor implementation (**e**) of the functional hypercycle (of 3 elements) ensures that all the functions will be optimally represented.

**Figure 8 life-10-00014-f008:**
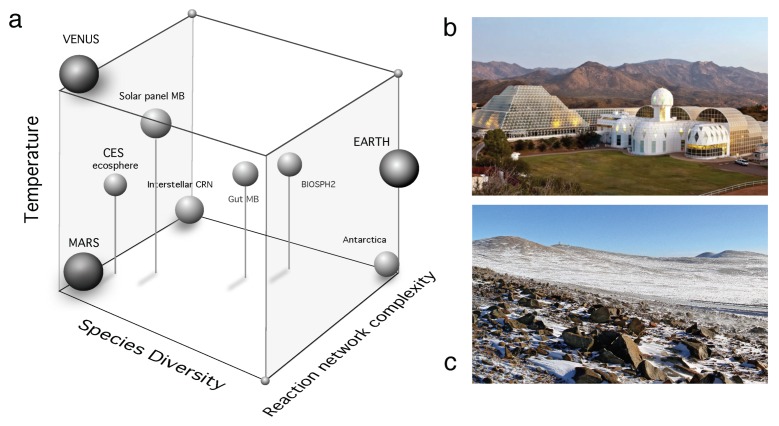
The space of the possible in Terraformation scenarios. In this space (**a**) we have placed (on a relative, qualitative basis) the different case studies relevant for the bioengineering designs that could be relevant for terraformation. The cube is defined by three axes including the diversity of living species, the complexity of the underlying web of chemical or biochemical reactions and the average temperature. Our planet and the two Earth-like counterparts (i.e., Mars and Venus) along with the complex chemical reaction network from Interstellar space, provide limit cases. Two examples of the systems including within this space are shown in the right panels: (**b**) the large-scale experiment Biosphere 2 and (**c**) the Atacama desert (images from *WikiCommons*).

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
