# Peer review of "Synthetic Biology for Terraformation Lessons from Mars, Earth, and the Microbiome"

_life, 2020, doi:10.3390/life10020014_

Round 1

Reviewer 1 Report

This article discusses the possibilities of engineering ecosystems to solve environmental problems. The authors show how mathematical models are used to analyze several case scenarios of ecological hypercycles, involving microbial communities. The interface of synthetic biology and ecology is a field of research that emerges rapidly in recent years. This article is expected to be interesting to both bioengineers and ecologists. However, a few issues must be addressed before this manuscript is ready for publication.

Figure 2 is not mentioned in the main text. This figure describes how cells can be modified to support the development of synthetic ecosystems, which is very important for readers to understand the basis of designing and building of terraformation motifs. A brief description of this figure should be added to the Introduction section. Readers from some fields, such as ecology and evolution, may not be familiar with the approaches used for engineering organisms. This reviewer appreciates that the authors have highlighted the cell circuit approach in Figure 2c. However, the information provided in that figure is not sufficient for some readers to understand how cellular behaviors can be modified. To fill this knowledge gap, a paragraph should be added to the Introduction section to briefly describe the engineering of signaling pathways that enable the communication between cells in a synthetic community. Some latest examples should also be presented to indicate the process of building microbial consortia at the molecular and genetic levels. For examples, the article, PMID: 27172092, illustrates the use of quorum sensing molecules to establish connections between cells; another article, PMID: 31162606, shows a circuit design that enables cells to memorize exposures to multiple environmental signals and use the information to control genetic activities; PMID: 29942078 describes the use of computational and experimental methods to establish microbial consortia with 2 to 4 species. These three papers highlight the engineering of circuit components, the design of cell circuits, and the development of synthetic ecosystems/microbial consortia. The authors may cite more studies to convince readers that we already possess sufficient synthetic biology techniques to pursue terraformation. Some figures and data are directly adapted from published papers. The authors may need to indicate that the use of these figures has been authorized by the corresponding publishers.

Author Response

We want to thank the referee for revising the article. Indeed the paper is at the interface of synthetic biology and ecology, we analyze several cases of ecological hypercycles of microbial communities. However, we did not provide details on how these synthetic approaches could be preformed at genetic level. We totally agree that we missed that point. and added a paragraph in the Introduction section.

As indicated by the reviewer, we now cite Figure 2 with emphasis on the genetic circuit approach of panel c and d.

The paragraph briefly describes the engineering of genetic circuits to respond to environmental signals, to control communication between microorganisms, to coordinate the behavior of microbial consortia and to create a synthetic microbial ecosystem. We thank the referee for the three informative references. We cited them in the text together with several classic works of synthetic biology and other more recent papers (from genetic techniques to engineering of synthropic exchanges of multiple strains consortia)

All adapted figures have been modified from the original published ones, and come from papers of the authors themselves.

Reviewer 2 Report

This engaging manuscript broadens the concept of using synthetic biology approaches to bioengineer ecosystems. The authors propose that there are significant connections between different types of systems, such as microbiomes, the Earth's biosphere, and the terraformation of other planets (e.g., Mars); and that the knowledge generated by studying any individual system can help to understand and engineer the others. Ultimately, this approach could enable us to reliably bioengineer ecosystems both at small (microbiome-level) and large (planet-level) scales.

I believe that there will be a significant need for terraforming research in the future, including terraforming Earth for restoring ecosystems, as the authors propose. Overall, I think that the manuscript is compelling, timely, and relevant.

Below I list many amendments to improve the manuscript before publication:

1) Page 1: “Given the accelerated growth of human population…”.

According to the United Nations, even though the world population is growing, its growth rate has halved from above 2% per year 50 years ago to ~1.05% per year (https://ourworldindata.org/world-population-growth). Therefore, I suggest the authors re-phrase this statement to, for example: “Given the continuous growth of human population…”.

2) Page 3: Please change “an habitable” to “a habitable”.

3) Page 3: Please change “a Earth-like” to “an Earth-like”.

4) Page 3: Please change “drylands as case study” to “drylands as a case study”.

5) Page 3: This sentence is hard to read: “However, to achieve the landscape-level or even planetary-level targets special classes of dynamical interactions, the so called ecological hypercycles, (defined as closed cooperative consortia) are required. Please change to “However, to achieve landscape-level or even planetary-level targets, special classes of dynamical interactions, the so-called ecological hypercycles (defined as closed cooperative consortia), are required.” or rewrite it.

6) Page 3: Please change “that can met” to “that can meet”.

7) Page 4: Please change “biocomplexity plants” to “biocomplexity, plants”.

8) Page 5: “This area promises the modification of living systems in ways that could overcome the design limitations resulting from evolutionary trade-offs.”.

Regarding the use of synthetic biology to overcome the evolutionary-derived limitations of Earth's living systems and allow for multi-planetary life, I believe that the authors should cite here relevant publications, including Verseux et al. Int J Astrobiol 2016, 15, 65–92 and Llorente et al. Genes 2018, 9:348.

9) Page 8: Please change “type) the surface” to “type), the surface”.

10) Figure 3, legend: Please amend “Bistablility” to “Bistability”.

11) Figure 3, legend: Please amend “Google Enginee” to “Google Earth Engine”.

12) Figure 3, legend: Please amend “displaied” to “displayed”.

13) Figure 3, legend: Please amend “Functionallity in semiard” to “Functionality in semiarid”.

14) Figure 3, legend: Please amend “coesxist” to “coexist”.

15) Page 8: Please change “cyclic we” to “cyclic, we”.

16) Page 9: Please change “conatrsint” to “constraint”.

17) Page 10: Please change “transition the become” to “transition they become”.

18) Figure 5b: I cannot distinguish a thin line and a thick line.

19) Figure 5b: Please ensure that the legends on the axes of the graph are readable.

20) Page 11: Please change “also facilitate optimal operation since, each” to “also facilitates optimal operation since each”.

21) Page 11: Please change “separated but interconnect” to “separate but interconnected”.

22) Figure 6, legend: Please change “shown in Fig. (5)(c)” to “shown in Fig. 5c”.

23) Figure 6, legend: Please change “specie” to “species”.

24) Figure 6, legend: Please change “system live” to “system to live”.

25) Page 12: I suggest changing “of a. a. for” to “of amino acids for”.

26) Page 12: Please change “reality , as” to “reality, as”.

27) Page 12: Please change “examples of fig. 6a.” to “examples of fig. 6.”.

28) Page 12: Please change “e.coli” to “Escherichia coli” as it is mentioned here for the first time in the text.

29) Page 12: I suggest changing “where cocultured together” to “where cultured together” or “where cocultured”.

30) Page 12: Please change “as summarized in (Fig. 7a)” to “as summarized in Fig. 7a”. 

31) Page 13: Please change “abundancea” to “abundance”.

32) Figure 7, legend: Please amend “First, if the synthetic organisms are deployed in the planet, depending on their death and the growth rates they will grow explosively or get extinct,” to “First, if the synthetic organisms are deployed on the planet, depending on their death and growth rates, they will grow explosively or get extinct.”.

33) Page 14: Please change “GE organisms” to “genetically engineered organisms” as it is mentioned here for the first time in the text.

34) Page 14: I suggest changing “become of essence for planning spreading” to “become important for planning the spreading”.

35) Page 14: Please change “…genetic designs remains” to “…genetic designs remain”.

36) Page 14: Please change “required human life” to “required for human life”.

37) Page 14: Please change “are nos separated” to “are not separated”.

38) Page 14: Please change “It a more broad sense” to “In a more broad sense”.

39) Page 14: Please change “is a second axes” to “is a second axis”.

40) Page 14: Please change “mentioning, One” to “mentioning. One”.

41) Page 15: Please change “inspiration of future” to “inspiration for future”.

42) Page 15: Please change “a no less important lessons” to “a no less important lesson”.

43) Page 15: Please change “for the future will immense” to “for the future will be immense”.

Throughout the manuscript:

44) For consistency, please change “i.e.” to “i.e.,”.

45) Please use one consistency: figure, Fig., Fig, fig or fig.

46) Please change “in Mars” to “on Mars”.

47) I suggest changing “aminoacid/s” to “amino acid/s”.

48) Please, double-check punctuation and for additional typos thoroughly.

Author Response

We want to thank the referee for his/her time and thoughtful review. We really appreciate the positive feedback and constructive comments, especially concerning our spelling, punctuation and grammatical errors. We have changed them all as stated by the referee. In general, we performed a more thorough proofread to correct the remaining typos and punctuation.

Also, as pointed out by the referee we have modified figure 5b, making thicker lines and increasing the axes legends to be more readable. We have also revised the text for consistency at citing the figures or using "i.e".

In this reviewed version of the manuscript, we added a paragraph at the end of the Introduction section (page 3) to explain better the genetic engineering involved in the proposed terraforming framework.

Round 2

Reviewer 1 Report

The authors have addressed all concerns of this reviewer.